# The Consensus of Global Teaching Evaluation Systems under a Sustainable Development Perspective

Yi Zhang [1,2], Siyu Sun [3], Yuhan Ji [4] and Yazhi Li [1,*]

1 School of Mathematics and Statistics, Qiannan Normal University for Nationalities, Duyun 558000, China
2 School of Mathematical Sciences, East China Normal University, Shanghai 200241, China
3 College of Elementary Education, Capital Normal University, Beijing 100048, China
4 College of Education, University of Washington, Seattle, WA 98195, USA
* Correspondence: liyz@sgmtu.edu.cn

**Abstract:** As an elemental driving force for promoting teaching reform, teaching evaluation has been receiving extensive attention in the fundamental reform of the overall education system. Using six dimensions including evaluation indicators, evaluation objectives, evaluation methods, interest relations, rights and roles, and accountability models, this article conducts a survey of eighteen teaching evaluation systems in seven countries including the United States, Germany, China, Japan, Australia, Singapore, and Chile. After analyzing these evaluation systems, this paper concluded the following major trends: evaluation indicators tend to be more standardized, evaluation objectives are closer to teachers' professional development, evaluation methods pay more attention to formative evaluation, interest relationships tend to be low-risk, and evaluation plays a diagnostic role in teachers' growth and the increased autonomy of schools in the accountability system. At last, this paper proposed that the future teaching evaluation system should focus on improving teachers' skills and profession, designing the evaluation system with the principle of combining practice and theory, and finally changing from high-risk summative results to low-risk formative ones. Through the above revelations, we hope to help educational policymakers systematically consider and solve core problems in the teacher evaluation system.

**Keywords:** teacher professional development; teaching evaluation system; comparison among countries

## 1. Introduction

Since the 20th century, the human mind, including pedagogical thinking, has been "moving from traditional substantive thinking that questions the origin or origin of the world and things to practical thinking that emphasizes the dominance of the subject and its practice in relationships" [1]. The French philosopher Augustin Auguste Comte, known as the "father of sociology", divided the human mind into three stages: theological, metaphysical, and empirical, that is, the "three-stage rule" of the human mind development. The empirical stage, mainly characterized by scientific observation and reasonable prediction, is the most advanced stage of human social development [2]. The development of practical thinking, especially the rise of empirical research, has provided a strong guarantee for human science to explore the laws of nature and social phenomena. Based on this, education researchers believe that the achievement of learners and social behavior can be predicted using systematic classroom observation and analysis. The paradigm of classroom research has also shifted from purely theoretical research to an open-field style that focuses on teaching practice [3]. In addition to the shift in the human mind and research paradigm, how to perform classroom teaching evaluation is also changing. Taking the United States as an example, the conventional teacher evaluation method is a comprehensive method developed by regions and states to evaluate teacher efficiency under the requirements of federal guidelines. The evaluation method is discussed and developed by educators,

statisticians, and education policymakers to evaluate teachers based on the total amount of student achievement, so it is called the value-added model (VAM) [4]. However, research shows that student achievement is rarely a driver of teacher evaluation [5]. Generally, classroom implementation based on classroom observation has greater weight. The common assumption in most teacher evaluation systems around the world is that classroom practice is a critical mediator between educational policy and student achievement [6], while classroom observation remains the best choice for understanding these teaching practices in a natural environment [7]. Observation-based teacher classroom evaluation is seen as key to understanding the mechanisms underlying classroom teaching and student achievement improvement. It also provides a starting point for guiding and improving teaching by forming and developing classroom teaching feedback [8]. To sum up, classroom teaching evaluation has shifted from focusing on static value-added models to emphasizing generative classroom practice observations.

At the beginning of the last century, educational research also began to draw on the research paradigm of natural science. The empirical research method was accepted by more and more scholars. Classroom research also paid attention to systematic and quantitative observation methods. Especially with the rapid development of education and teaching reform, directly observing the classroom has become the main way to study teacher education and teaching work. The study of classroom teaching evaluation using classroom observation has become a hot spot in international education research [9]. Validation research is once again underway as a new generation of researchers works to develop observational protocols and tools to help understand the relationship between classroom practice, teacher effectiveness, and student achievement [10]. Classroom teaching evaluation varies according to the specific system and environment, but every evaluation system has a clear basic structure and mechanism. The primary problem in teaching evaluation systems design is establishing theories, clarifying concepts, and then forming a basic framework. To better achieve this, many education systems are standardizing their approach to classroom observation to make them consistent with the framework. Expect unambiguous teaching standards, matching observation rules can provide useful guidance for teachers and managers to understand and promote high-quality teaching [11]. The mechanism of classroom teaching evaluation further focuses on the roles of different groups in the evaluation process, the interests generated using the evaluation results, and the accountability model after the evaluation. However, facing multiple teaching evaluation systems, we need to carefully consider how to use them to interpret the classroom. How educators and policymakers use the system to collect classroom information and how to use it to give teacher feedback is worth studying.

Based on the above discussion, we proposed the research question: what are the consensuses of the teaching evaluation systems currently used worldwide? This research analyzes how the education systems in different countries and regions of the world apply classroom observation to guide teacher evaluation and professional development. The international comparison method is used to summarize and analyze the basic conceptual issues and policy issues including six dimensions of evaluation indicators, evaluation objectives, evaluation methods, interest relations, rights and roles, and accountability model, so as to help decision-makers comprehensively consider the key issues involved in the design of the teaching evaluation system.

## 2. Theoretical Background

The concept of sustainable development is named after the Brundtland report, which reported sustainable consumption in developed countries. As an organizing principle for social development, it has developed specific definitions of these three fundamental pillars: social, economic, and environmental [12]. In education, for example, education for sustainable development is a term used by the United Nations and is defined as education that encourages changes in knowledge, skills, values, and attitudes to enable a more sustainable and just society for all [13]. Agenda 21 was the first international document

that identified education as an essential tool for achieving sustainable development and highlighted areas of action for education.

For UNESCO (The United Nations Educational, Scientific and Cultural Organization), education for sustainable development involves integrating key sustainable development issues into teaching and learning. It also requires participatory teaching and learning methods that motivate and empower learners to change their behaviors and take action for sustainable development. Education for sustainable development consequently promotes competencies like critical thinking, imagining future scenarios, and making decisions in a collaborative way [14]. One version of education for sustainable development recognizes modern-day environmental challenges and seeks to define new ways to adjust to a changing biosphere, as well as engage individuals to address societal issues that come with them [15].

Different evaluation systems differ in the assumptions of objectives and aims, the interest relationship and incentive measures of evaluation results, the framework construction of evaluation fields and criteria, the definition of models, and the methods of collecting and analyzing information [8]. However, from the sustainable development perspective, there are some consensuses to teaching evaluation systems in various countries. This study selected eighteen teaching evaluation systems in seven countries as a sample to explore these consensuses.

For a teaching evaluation system, the first step is to clarify the theoretical or conceptual underpinnings that will provide the basis for understanding, describing, and assessing teacher practice. Second, policy-related factors are as important as conceptual considerations for influencing the design, credibility, and sustainability of a teacher evaluation system [16]. Therefore, we outline an analytic framework for teaching evaluation systems that involves these two dimensions: conceptual issues and policy issues.

Conceptual issues can be summarized in three aspects: evaluation indicators, evaluation objectives, and evaluation methods. These are the main content of an evaluation system, constitute the material shell of evaluation systems, and become the premise of effective and credible evaluation systems.

The literature consistently suggests that contextual factors are particularly relevant where the political context is charged [17]. Here, we consider the interest relationship, rights role, and accountability model, which are three key policy factors that may inform and explain the conceptual and technical features of evaluation systems. These undertake the task of transforming material goals into specific practical goals, which constitutes the software environment of the teaching evaluation system and is a strong guarantee for the effective implementation of education and teaching evaluation.

## 3. Methods

### 3.1. Sample

We seek to demonstrate and describe the variety of uses and approaches to teaching evaluation in a variety of national, regional, and local contexts around the world. Thus, we selected 18 teaching evaluation systems in 7 countries worldwide as the sample. They are all widely used evaluation systems that are representative of those countries or regions. We purposively selected observation systems designed in the context of teacher professional development and appraisal, or school evaluation, for which we were able to obtain sufficiently detailed information from available documents, as shown in Table 1.

The sample includes the three largest school districts in the United States: New York, Los Angeles, and Chicago. In recent years, these districts have redesigned teacher evaluation systems to align with the America COMPETES Act of 2022 guidelines. These systems comprehensively integrate students' academic growth, teachers' practical performance, and other indicators [18]. Tennessee did not add student indicators to the system but widely applied the National Institute for Excellence in Education's TAP[TM] model [19]. The evaluation systems of Cincinnati and Toledo rely on classroom observation and focus on peers, experts, and administrators, serving as high-risk summative evaluation models for teacher promotion and recruitment [20]. Although this model does not include student

achievement as a factor in high-risk assessment, the research suggests that the system's observation score is highly correlated with the value-added index of teacher effectiveness [21]. This is because the observation process leads to an increase in the teacher's value-added score. Similarly, Pittsburgh's evaluation system incorporates a formative assessment of classroom observation into teacher training, promotion, and other closely related interests for teachers [22]. The Santa Monica Unified School District system embodies a common model for small- and medium-sized regions in the United States, with schools having considerable discretion and detailed, standardized observations that provide timely feedback to teachers [23]. For example, the National Board of Professional Teaching Standards (NBPTS) provides teachers with evaluation standards for different grades and different subjects and identifies the best teachers according to the standards [24]. The leadership performance teaching system emphasizes the student population to improve teacher pass rates. Math Scan (M-Scan) is an assessment tool developed specifically for the practice of teaching mathematics and epitomizes the Common Core State Mathematics Curriculum Standards and the Standards and Principles of Mathematics Education in Schools [25].

**Table 1.** Information on the International Teaching Evaluation System.

| System | Country (Region) | System | Country (Region) |
|---|---|---|---|
| Teaching Seminars | Australia (Victoria) | Math Scan (M-Scan) | USA |
| LICC mode | China | Teaching Excellence | USA (Chicago) |
| Teacher Professional Performance Evaluation System | Chile | Teacher Evaluation System | USA (Cincinnati) |
| School Teacher Assessment System | Germany (Hamburg) | The Educator Growth and Development Cycle | USA (Los Angeles) |
| External Assessment System | Germany (Saxony) | Annual Professional Performance Appraisal | USA (New York) |
| Performance Evaluation System | Japan (Fukuoka) | The Research-based inclusive Assessment System | USA (Pittsburgh) |
| Enhanced Performance Management System | Singapore | The Standards-based Teacher Evaluation System | USA (Santa Monica) |
| National Council on Professional Teaching Standards | USA | Tennessee Educator Acceleration Model | USA (Tennessee) |
| Teaching As Leadership performance teaching system (Teach for America, TFA) | USA | Toledo Internship Program | USA (Toledo) |

In China, there are many studies on classroom teaching evaluation, but there are few standards-based professional teaching evaluation systems. In this study, the professional lecture mode, which has a certain influence in China and has a wide range of applications, is selected, which is called the classroom observation LICC paradigm. The system consists of "Learning", "Instruction", "Curriculum", and "Culture", which can be refined into 20 perspectives and 68 observation points [26]. Chile's teacher professional performance evaluation system collects teacher documents, which include written evidence of work and videos of a lesson, as well as supervisor questionnaires, peer interviews, and self-assessments, and formative feedback on teachers relies heavily on professional video lessons [27]. In Australia, the Victorian Teaching Seminar has been chosen, which is an evaluation system that supports teacher improvement in a direct and contextualized manner, primarily in a team form, using regular classroom observations [28]. Singapore's education is known for the high level of achievement of its students in international assessments, and its teacher evaluation methods are distinguished by their low risk and dual focus on excellence and professional development. Its Enhanced Performance Management System (EPMS) serves as the primary reference model for classroom practice and teacher

competence [29]. In Germany, teaching evaluation is used for school evaluations or inspections. This approach relies on systematic evaluation criteria, such as the Hamburg School Teaching Assessment System and the Texas State External Assessment System, which were selected for this study, and both have clear evaluation standards [30].

*3.2. Measures*

According to the theoretical framework, we analyze the samples under six dimensions: evaluation indicators, evaluation objectives, evaluation methods, interest relationship, rights role, and accountability model.

Evaluation indicators are the specific, observable, and measurable points declared in the systems. They are usually stated explicitly in the framework of the system. The evaluation objective is a focus and wrap of the goals of the system in measurable outcomes. The evaluation method reflects the system's emphasis on formative evaluation or summative evaluation, and our analysis also focuses on these two points. Furthermore, interest relationship, rights role, and accountability model are three dimensions that are the main components of political context. They are always described in the introduction documents of most evaluation systems. Interest relationship is the stakes associated with the evaluation for teachers, such as the impact of the evaluation results. Rights role especially means the locus of control for the evaluation or the degree of local discretion. Accountability models focus on the type of accountability model underlying the approach to teacher evaluation.

*3.3. Method for Analysis*

International comparative research is widely employed to describe studies of societies, countries, cultures, systems, institutions, social structures, and changes over time and space [31]. In the education community, it emphasizes the value of analyzing similar experiences across different contexts to derive insights that can inform educational practice and policymaking. This study aims to understand and classify the rising consensuses of existing systems of teaching evaluation in educational jurisdictions around the world, along dimensions of conceptual and policy variation. However, our analysis does not focus on comparison but seeks to outline an analytic framework of teaching evaluation systems based on conceptual and contextual commonalities and differences. The analytic approach involved qualitative examination and synthesis of information from multiple sources (publicly available documents). We triangulate or "cross-examine" the initial descriptions gained from publicly available documents by interviewing some related district/system personnel, and by examining other related documents. Then, we organized the analysis around the analytic framework, which comprises conceptual issues (evaluation indicators, evaluation objectives, and evaluation methods) and policy issues (interest relationship, rights role, and accountability model). These dimensions were the basis for classifying systems and highlighting areas of conceptual commonality and policy contexts.

## 4. Findings

### 4.1. Evaluation Indicators Tend to Be Standardized

It is generally believed that there are two different models of instructional evaluation: the standards-based model and the outcome-based model. The standards-based model emphasizes a clear analytical framework to assess the quality of teaching and further guide classroom teaching practice [32]. Outcome-based models use student achievement and other relevant outcomes to measure the output of a teacher's teaching [7]. The two models have different characteristics. The standards-based model usually focuses on uniformity of evaluation, emphasizing teaching coverage and capacity. Outcome-based models place greater emphasis on achieving instructional goals and increasing student performance. The recent trend tends to converge the two models, combining student achievement or other relevant outcomes with clear and detailed models of teaching practice [33].

After summarizing 18 evaluation systems, it was found that almost all systems focus on the standardization of evaluation indicators. Standardized evaluation indicators reduce

the quality requirements of evaluators to a certain extent and increase the consistency of evaluation. Especially in recent years, the implementation of curriculum standards in various countries has formed a standardization-oriented educational development model, and the formulation of standards and the implementation of teaching using standards have become the main forms of teaching development. Observation systems such as those developed by different districts in the U.S. are designed to standardize processes as much as possible to reduce discretion [21]. The standardization of the teaching evaluation system is essentially influenced by curriculum standardization. With the introduction of curriculum standards in various countries, teaching content arrangement, teaching process design, and student academic evaluation are more standardized. In this analysis, it was found that all systems evaluate teachers' knowledge of the content, as well as their ability to plan and set instructional goals. At the same time, each model considers the student's characteristics and applies appropriate assessment methods in the classroom as a guarantee for high-quality teaching [34]. Moreover, the standardized design of teaching evaluation indicators also provides a premise for the automatic evaluation of teaching and more reasonable data processing. Standardized assessment has also led to the development of various computer-adaptive assessment software, which makes teaching evaluation more intelligent.

### 4.2. Evaluation Objectives Point to Teachers' Professional Development

Every evaluation activity involves its value orientation, and teaching evaluations are no exception. The goals of teaching evaluation ultimately reflect the value orientation of the evaluation system. Evaluation objectives are often reflected in specific indicators. In these evaluation systems, the goal of evaluation is mainly reflected through information such as the frequency of evaluation, the object of evaluation, and who uses the evaluation results. By analyzing these 18 evaluation systems, it was found that if the evaluation system is used to monitor routine teaching, the evaluation is generally for all teachers, and the evaluation can be carried out once or more times a year. If the purpose of the evaluation is to understand the case in-depth, the assessment will be carried out several times from time to time. Some results of the assessment are used for teacher selection and promotion, but overall, most of the 18 assessment systems focus on the professional development of teachers. That is, the purpose of the evaluation is to further improve the teaching of teachers and enhance their teaching professional competence. For example, in the framework of the National Council on Professional Pedagogical Standards, the assessment focuses more on pedagogical technology and procedural capabilities, such as teachers' questioning skills, classroom management skills, and the ability to organize students to cooperate in small groups [5]. In Singapore's system, teaching assessment is more focused on the achievement of pedagogical objectives. The results of the assessment will be fed back to the evaluated teacher promptly, and an in-depth discussion will be conducted with the teacher to improve their teaching skills. At the same time, Singapore's teaching assessment system pays special attention to the cultivation of students' interests, including guiding students to build self-confidence, establishing correct values, and sharing the joy of growth between students and teachers [35]. By comparison, the Chinese evaluation system places more emphasis on teaching rules and monitoring. The goal of the evaluation focuses on the student's performance and the teacher's teaching behavior, specifically, whether the teacher's expression is fluent and whether the teaching is implemented according to the predetermined plan. The system pays little attention to the learning opportunities of students or the physical and mental health development of students. To sum up, it has become a consensus that teachers' professional development is the goal of the teaching evaluation system.

### 4.3. Evaluation Methods Focus on Formative Results

Although there are great differences in the specific assessment method of different evaluation systems, in recent years, the research on teaching evaluation in various coun-

tries has shown a common feature—focusing on a formative evaluation. For example, the federal election bill requires results of the system data to "inform teachers and principals how they can improve teaching" [36]. While the National Teacher Performance Agreement in Australia aims to promote professional dialogue and improve teaching [37]. Similarly, Chilean legislation establishes a formative teacher evaluation system that focuses on "improving the teaching and learning of teachers and promoting their continuous professional development" [38]. After each evaluation, the evaluator will provide a detailed report to the participating teachers and provide a reference for those teachers to self-diagnose and improve their teaching skills [39]. Influenced by these policies, the teacher's teaching evaluation system pays more attention to formative evaluation. The American As Leadership Performance Teaching System (TFA) only uses formative assessment results, that is, to improve teaching with the goal of increasing students' interest [40]. For the systems in Singapore, Germany, and Santa Monica, the information collected through observation is primarily used for formative evaluation to improve teachers' teaching practices [41]. The reason why formative assessment is so important is closely related to the characteristics of classroom teaching activities. A classroom is a place where thinking is constantly changing and situations are constantly generated, and the continuous change of teaching determines that the evaluation of classroom teaching should not only focus on the results but should pay attention to the process. It can be seen that the summative evaluation of teachers through a lesson or some teaching fragments has gradually been replaced with formative evaluation, which can help teachers improve teaching more comprehensively.

### 4.4. Teachers' Risks Reduce in Evaluation

Based on the assessment and the different uses of the evaluation results, the impact on the teacher being evaluated is called the evaluation risk. Among the 18 evaluation systems, most of the evaluations are used to further help teachers maintain good teaching, improve the problems encountered by teachers in the teaching process, and then improve the overall teaching quality. For example, the Chicago and Tennessee systems differ markedly in the assessment of novice and tenured teachers, who tend to be observed more frequently and take higher risks. Furthermore, in the evaluation systems of Toledo, Santa Monica, and New York, new teachers are at higher risk of making tenure decisions [42]. In addition, some evaluation systems use the results of multiple classroom observations to evaluate teaching, which reduces the risk of the teacher being evaluated. For example, the evaluation system in Pittsburgh, USA, evaluates teachers several times a year and then discusses them with peer teachers to support the teachers' skill improvement, which does not bring evaluation pressure to teachers, and teachers have a low risk of evaluation [24]. The evaluation systems in Singapore, Germany, and Santa Monica focus on the interests of teachers and students and improve teaching in ways that are of interest to teachers and students. The consequences for teachers are non-punitive, with a tendency to improve teachers' teaching practices, and the low risks prompt teachers to improve teaching [43]. The case of Singapore's evaluation system shows that the lower the risk of punishment, the more positive the impact on teachers [29]. Most evaluation systems have both formative and summative assessment functions. Formative assessment is more conducive to being accepted by teachers, thereby reducing the risk of evaluation, and then urging teachers to carry out teaching reform, to improve the level of teacher professional development.

### 4.5. The Evaluation System Functions as a Diagnosis

The operation mechanism of the evaluation system is the fundamental guarantee of the effective operation of the evaluation system. Through the analysis of 18 evaluation systems, it was found that most of the evaluation systems will be fed back to teachers in different ways to help the teachers perform self-reflection and adjustment. The diagnostic function of the evaluation system in diagnosing teachers' teaching behavior has reached a consensus among the evaluation systems, but the way to give feedback information is purely different. In Chile's evaluation system, for example, evaluators generate feedback

based on teachers' written materials (lesson plans, notes, homework feedback, etc.) and classroom videos, and then offer teachers written reports detailing their behavior on seven different dimensions, and they also use the results of the evaluation to determine the allocation of federal funds for teacher professional development [39]. In Germany., the evaluators can provide feedback to the school committee, propose the issues observed in the classroom assessment, and then the principal can discuss with the teacher to map out the school's teacher professional development plan [43]. In Tennessee, Chicago, and Los Angeles, before the assessment, the teacher being evaluated will discuss the focus of the assessment. After the evaluation is completed, the evaluation results will be given to the evaluated teachers to further discuss the improvement plan, including the future development direction [5]. In Toledo's evaluation system, a peer-to-peer model is used, in which new teachers and expert teachers pair up to help new teachers grow based on the information collected during teaching [5]. In conclusion, the diagnostic function of the evaluation system is already one of its most important functions and has been highlighted in many evaluation systems.

### 4.6. School Autonomy Increases in Accountability

The accountability model involves issues such as who has the authority to conduct the assessment, who uses the results of the assessment, and who is the subject of the assessment. Generally, the accountability model is divided into two types: the professional model and the organizational model. The low degree of standardization of the professional model, which gives the evaluator greater ownership, tends to produce evaluations with lower punitive benefits, often based on the flexibility of the professional group to provide accountability, as in the Toledo state in U.S. and Singapore systems [44]. In contrast, the organizational model provides standardized accountability results as much as possible, reducing the discretion of the evaluator to some extent [21]. This is evident in observation systems developed by university districts in the United States. Organizational models increase accountability to some extent and can lead to more systematic improvements. Overall, schools play a more important role in accountability and have greater autonomy. For example, in Japan's evaluation system, principals can freely collect data and information using standardized observations during the evaluation process to determine the teachers' plans and needs for future professional development [26]. In New York City, however, evaluators, including principals, must be trained before the assessment and pass an online test to participate in the assessment. Delegating assessment tasks to a third party is also known as the main way of instructional assessment. The Chilean Ministry of Education entrusts the evaluation to the University Evaluation Centre. During the evaluation process, the Evaluation Committee may participate in the supervision and has the power to approve and modify the evaluation results. In Australia, different schools have developed specific classroom observation programs, so-called "teaching rounds", to further improve overall education. In this way, schools can freely decide the method, purpose, use of observations, and autonomy of evaluation. The initiative of evaluation is handed over to the school, and the process and results of the evaluation are at the discretion of the school, which improves the autonomy of education to a certain extent.

## 5. Discussion

Based on the concept of sustainable development of the teaching evaluation system, this study sorted out the basic framework of 18 global evaluation systems in detail. On the basis of comparative analysis, six trends of the global teaching Evaluation system are extracted, namely: the evaluation indicators tend to be standardized; the assessment objectives are directed toward the professional development of teachers; assessment methods focus on formative evaluation; the teachers' risk reduction in assessment; the evaluation system plays a diagnostic function; and the increased autonomy in accountability. Summarizing these 6 trends, the following three aspects are discussed in depth:

### 5.1. The Direction of the Evaluation System: Focusing on Teacher Skills Improvement and Teacher Professional Development

Although the teaching evaluation system is very different in terms of evaluation dimensions and methods, most of them provide feedback on the evaluation results to teachers. For example, after the completion of each evaluation, the Chilean teaching evaluation system will provide a detailed report card to the participating teachers and feedback on the results of the teacher evaluation to achieve the purpose of teacher self-diagnosis and improvement of teaching [39]. The assessment system in Pittsburgh, USA, evaluates teachers several times a year and then supports teacher skills improvement using peer–teacher discussions. For most teachers, there is a desire and space for self-improvement, but most teachers have a lack of support systems for self-reflection and improvement. German educator Schneider's research on self-cognition shows that individuals reconstruct themselves because there is a gap between the "real image" and the "ideal image". One's view of the self and of the world is present as an image. The "real image" presents the self, while the "ideal image" presents the world. The former is always immature, and there is a gap between them. Therefore, constantly deconstructing the "real image" and chasing the "ideal image" has become the normal state of each individual [45]. This can be understood as the original motivation for teachers' upskilling and professional development, but this motivation requires a relatively stable support system if it is to have sustained energy and the right direction. A good teaching evaluation system can provide teachers with powerful tools to identify problems, diagnose teaching, and support the improvement of teachers' teaching skills. Comparing the observation objects and frequency, this paper discusses that in addition to the evaluation system for promotion and recruitment, more evaluation systems are aimed at all teachers' development and pay attention to lifelong evaluation. More macro- and long-term evaluation data can provide a clearer direction and give specific measures for teachers' professional development. At present, China is still in the initial stage of utilizing evaluation to promote teachers' teaching skills, and most of them still use the subjective suggestions of experts to improve. Most of the support for teachers' professional development is in the form of training, but this method is not very targeted, is not sufficiently operable, and thus has little effect. Only by providing long-term support for teachers in a timely and accurate manner can teaching evaluation ensure the steady improvement of teachers' teaching quality. Therefore, making teaching assessment tools serve teachers' teaching skills improvement and professional development is the basic concept of system development.

### 5.2. Approaches to Designing the Evaluation Systems: Combining Classroom Practices and Theories

The structure of teacher evaluation systems depends on local contexts and cultures, as do their evaluation mechanisms and review processes. However, there are commonalities in the core aspects of the evaluation system. For example, the student-oriented nature of the evaluation system, the teaching effect and teaching skills, and the professional responsibilities of teachers. Although commonalities can be found in evaluation systems, due to the complexity of their infrastructure, the different focuses and operations of these common dimensions leave more room for the design of teaching evaluation systems. For example, the American system tends to focus on teachers' behavior, generally focusing on the detailed aspects of teaching practice and teaching content, and then evaluating it using empirical data linking individual practice to student achievement. In contrast, Singapore's system focuses on more general dimensions such as creative teaching, which are key indicators recognized by peers and experts by inducting the pedagogical characteristics of teachers [5]. The TFA framework balances the management of classroom behaviors by considering students' psychological characteristics. It was derived from an in-depth study of the teaching behaviors that most significantly improve student achievement [46].

Among the many evaluation systems, the American teaching practice evaluation system leads the development direction of the entire evaluation system with its pluralism

and diversity. One of its biggest characteristics is to emphasize the importance of grounded theory, building an assessment framework based on solid theory, ensuring the qualifications of evaluators, and ensuring the credibility and validity of assessment to the greatest extent. Second, teaching evaluation is based on the concept of teaching for learning, and using empirical research, effective classroom teaching behaviors are identified, and then the assessment framework is optimized to better play the role of evaluation in diagnosing and promoting teaching [47]. "Danielson's team developed a teaching evaluation system based on constructivist pedagogical theories based on the American Alliance for New Teacher Evaluation and Support's teacher professional development standards [48]. Rooted in psychological theories such as attachment theory and self-determination theory, Pianta's team has developed an evaluation system consisting of three modules: teacher emotional support, classroom organization, and teaching support [49].

At present, teachers have made great achievements in practice, and most teaching evaluations only stay in the evaluation of the experience and have not risen to a certain theoretical height, resulting in a lack of professionalism in teaching evaluation. It is difficult to form a systematic evaluation plan based on the evaluation results to guide teaching. Evaluation has also become a simple exchange of experience, which is difficult to rise to the theoretical level and form a systematic evaluation. However, teaching evaluation should be a comprehensive and systematic diagnostic process with a solid theoretical foundation, and the results of the evaluation can form a teaching improvement plan and provide actionable improvement measures for teaching. It can be seen that the teaching evaluation system is a widely used evaluation model derived from classroom teaching practice and rooted in a profound theoretical foundation. Classroom teaching practice solves the problem of individuality in system development, making the system more in line with the local background and culture. Rooting in the theoretical foundation solves the common problems of system development, making the system more universal and far-reaching.

*5.3. Evaluation System Outcomes: A Transition from High-Stakes Summative to Low-Stakes Formative*

There is a basic consensus on the importance of formative assessment in promoting student learning [50]. The fundamental purpose of teaching evaluation is to better promote teaching and further help the professional development of teachers. This purpose-determined teaching assessment will inevitably transform from high-risk summative to low-risk formative. However, based on the above research, especially the analysis of the "interest relationship" in the evaluation system, it is clear that the evaluation of teachers' teaching quality has also shifted from high-risk summative assessment to low-risk formative assessment. Of the 18 evaluation systems, most of them are aimed at low-risk and contribute to the improvement of teachers' professional competence, and only a few of the results of the system are used for high-risk purposes for teacher promotion and recruitment.

It can be seen that the orientation of the entire teaching evaluation system is the formative evaluation of teachers, and the goal of the evaluation should be to diagnose teachers' teaching problems, lead the improvement of teachers' education and teaching ability, and clarify the direction of teachers' professional development. The position of evaluation should take teachers as the main body of value and put the development of teachers in the first position of evaluation. The evaluation mechanism serves teachers to the greatest extent, and the accountability model cannot discourage teachers' enthusiasm for development, fully highlighting the important position of teachers in the teaching evaluation process. The evaluation of teachers' teaching under the guidance of formative assessment can make teachers easily show their real performance in the evaluation process and can reflect the problems existing in teaching to the greatest extent. This requires reducing the entanglement of teachers' interests in the evaluation process and focusing on the improvement of classroom teaching quality and teacher professional development,

which is not only beneficial to the teacher participating in the evaluation but also beneficial to the student in the classroom.

While it is theoretically logical to combine formative and summative assessments in the same evaluation system, how the two evaluation modalities can be successfully parallelized in practice remains an open question. In particular, formative assessment is difficult to implement without the help of other technical means, relying only on ordinary classroom observations. Therefore, to achieve the formative assessment, one also needs to use information technology and big data analysis and collect multi-classroom information for a more comprehensive evaluation [16]. While formative evaluation is important, summative evaluation is also an indispensable part of the overall education system, and sometimes the results of the summative evaluation must be used for teacher promotion and job evaluation. The exact data from formative evaluations can provide more reliable support for summative evaluation, make the results of summative evaluation more objective and reliable, and thus achieve internal unity.

In conclusion, some consensuses under the perspective of sustainable development are rising in the global teaching evaluation systems. The role of promoting formative evaluation and improving teacher professional development is more prominent, while the risk in evaluation is decreasing. This study also has some limitations. First, the sample selection is representative but not comprehensive enough. Evaluation systems within some other large cultural and economic regions such as Oceania and African countries were not selected for the sample. The findings of this study were only obtained from selected samples. Second, this study only analyzes the documents of the teaching evaluation system, which does not include their realistic implementation. Further follow-up investigation on the implementation of the teaching system can help to further understand and explain the research findings in this paper, and also make the conclusion more convincing.

**Author Contributions:** Y.Z. performed the data collection and analysis and completed the paper; S.S. took charge of the language work and proofread the article; Y.J. made a complete correction to the language of the article; Y.L. gave some good advice and provided Funding support. All authors have read and agreed to the published version of the manuscript.

**Funding:** This work was supported by the 2021 Humanities and Social Science Research project of the Ministry of Education of China under Grant No. 21YJC880102; the Guizhou Educational Science Planning Project under Grant No. 2021C046; the Education planning project in Qiannan Prefecture (2018A038); the Program of Qiannan Normal University for Nationalities under Grant Nos. (qnsy2019005, qnsy2019006).

**Institutional Review Board Statement:** This study does not contain identifying information, so it did not require ethical approval (Committee on Human Research Protection of ECNU. http://life.ecnu.edu.cn/_upload/article/files/de/d9/250ac9ff4d2eb27003ff30bb0269/c4ab83b2-ac10-454d-a352-715b70bb92e4.pdf (accessed on 12 October 2022)). No animal studies are presented in this manuscript. No potentially identifiable human images or data is presented in this study.

**Informed Consent Statement:** Not applicable.

**Data Availability Statement:** The datasets used and/or analyzed during the current study are available from the corresponding author upon reasonable request.

**Conflicts of Interest:** The authors declare no conflict of interest.

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
