# Peer review of "The Consensus of Global Teaching Evaluation Systems under a Sustainable Development Perspective"

_sustainability, doi:10.3390/su15010818_

Round 1
Reviewer 1 Report
1. The article compared and analyzed 18 teacher-teaching evaluation systems in seven countries through a sustainable development perspective using six dimensions of evaluation indexes. Please elaborate and clarify the key words (concepts) such as sustainable development perspective in this article. The authors have to provide good rationales for why these particular 18 educational systems in those seven countries (any criteria?) as well as theoretical foundations of the dimensions of evaluation indexes.
2. On page 3 line 102, It is suggested to elaborate COMPETES guideline when it first appears in the article so that the readers can follow the idea of teacher evaluation system. For example, what does the acronym of “COMPETES” stand for?
3. In the “research objects and their characteristics” section, authors listed 18 teacher-teaching evaluation system in seven representative countries by a table and briefly introduce each system in paragraphs. However, it would be clearer a for readers by combining the table and introductions of each system. For example, the table could have three columns with system, introduction/dimensions, and country(region).
4. There is a lack of rationales or a theoretical framework to use these six dimensions to evaluate teacher-teaching evaluation system and how these six dimensions are used to analyze the system. I would suggest having a data analysis section before the “findings” section or adding paragraphs in the research objects and their characteristics” section to have the rationales or certain theoretical framework of using the six dimensions: evaluation indicators, evaluation objectives, evaluation methods, interest relationships, rights roles, and accountability models. Most importantly, the analysis procedure should have been clearly laid out as to illustrate how the findings(results) were obtained.
5. There are several duplicate sentences on page 7-8, line 328-333. For example,
“…systems is extracted, namely, Evaluation indicators tend to be standardized; The assessment objectives are directed toward the professional development of teachers; The assessment objectives are directed toward the professional development of teachers….”
Please carefully proof read before submission.
6. In the discussion section, the authors could also have a paragraph to wrap up the findings and the conclusions of the study. What is the takeaway (implication) from the study?
7. The manuscript requires professional English editing.
Author Response
Q1: The article compared and analyzed 18 teacher-teaching evaluation systems in seven countries through a sustainable development perspective using six dimensions of evaluation indexes. Please elaborate and clarify the keywords (concepts) such as sustainable development perspective in this article. The authors have to provide good rationales for why these particular 18 educational systems in those seven countries (any criteria?) as well as theoretical foundations of the dimensions of evaluation indexes.
Answer: We have elaborated and clarified the keywords such as sustainable development and given an explanation of why these evaluation systems were selected in the theoretical and methods section.
Q2 On page 3 line 102, It is suggested to elaborate COMPETES guideline when it first appears in the article so that the readers can follow the idea of teacher evaluation system. For example, what does the acronym of “COMPETES” stand for?
Answer: COMPETES means the America COMPETES Act of 2022. We have modified it to make it clear to understand.
Q3. In the “research objects and their characteristics” section, authors listed 18 teacher-teaching evaluation system in seven representative countries by a table and briefly introduce each system in paragraphs. However, it would be clearer for readers by combining the table and introductions of each system. For example, the table could have three columns with system, introduction/dimensions, and country (region).
Answer: It is a good idea to combine the introduction paragraph of each system to the table. We tried but failed. The introduction of each system includes different aspects, it makes the table incongruous and lengthy. From the other side, some comparison and detailed explanation have been made in the paragraphs, which are hard to add into the table. So, we still keep the table and the paragraph separately.
Q4 There is a lack of rationales or a theoretical framework to use these six dimensions to evaluate teacher-teaching evaluation system and how these six dimensions are used to analyze the system. I would suggest having a data analysis section before the “findings” section or adding paragraphs in the research objects and their characteristics” section to have the rationales or certain theoretical framework of using the six dimensions: evaluation indicators, evaluation objectives, evaluation methods, interest relationships, rights roles, and accountability models. Most importantly, the analysis procedure should have been clearly laid out as to illustrate how the findings(results) were obtained.
Answer: We have added a theoretical background section to give some rationales for these six dimensions. And we also added a paragraph on the analysis approach to make the analysis process clear.
These six dimensions constitute two aspects: conceptual commonality and policy contexts. These two aspects together constitute the main component of a teaching evaluation system, and provide effective and credible evidence for analyzing the teaching evaluation system.
For the analysis approach, it’s a kind of international comparative research that doesn’t focus on the comparison but on outlining an analytic framework of teaching evaluation systems based on conceptual, and contextual commonalities and differences. The analytic approach involved qualitative examination and synthesis of information from multiple sources (publicly available documents). We triangulate or “cross-examine” the initial descriptions gained from publicly available documents by interviewing some related district/ system personnel, and by examining other related documents.
Q5. There are several duplicate sentences on page 7-8, line 328-333. For example,
“…systems is extracted, namely, Evaluation indicators tend to be standardized; The assessment objectives are directed toward the professional development of teachers; The assessment objectives are directed toward the professional development of teachers….”
Please carefully proof read before submission.
Answer: We have modified this mistake and carefully read the whole manuscript and have a detailed editing correction.
Q6. In the discussion section, the authors could also have a paragraph to wrap up the findings and the conclusions of the study. What is the takeaway (implication) from the study?
Answer: We have added a paragraph to wrap up the findings and the conclusions of the study.
Q7: The manuscript requires professional English editing.
Answer: A researcher professional in English has edited the manuscript.

Reviewer 2 Report
I enjoyed reading this interesting article.
It seems that deals with an important topic and represents a real problem in many educational systems worldwide, which is the subject of the teaching evaluation system.
I did not understand the methodology used, I did not find specific questions that the study answered, and since I could not define a specific goal For the study, there are no clear tools for me in collecting data, analyzing it, and the results reached. This study seems to me to be a mixture of theoretical review, oscillating here and there without a clear scientific methodology for me
suggestion:
If the researchers separate the study methodology and analysis section from the results review part with the appropriate citation of the analysis methodology used in the study.
Author Response
Reviewer #2:
I enjoyed reading this interesting article.
It seems that deals with an important topic and represents a real problem in many educational systems worldwide, which is the subject of the teaching evaluation system.
I did not understand the methodology used, I did not find specific questions that the study answered, and since I could not define a specific goal for the study, there are no clear tools for me in collecting data, analyzing it, and the results reached. This study seems to me to be a mixture of theoretical review, oscillating here and there without a clear scientific methodology for me.
Suggestion:
If the researchers separate the study methodology and analysis section from the results review part with the appropriate citation of the analysis methodology used in the study.
Answer: Thank you very much for your advice. We have changed the stated objectives to research questions in the introduction part. Besides, the methodology is also added following the theoretical part.

Round 2
Reviewer 1 Report
All the comments and issues are addressed except for the combination of each system detailed introduction into a table. However, the way that it is presented with a concise table and paragraphs separate is acceptable as well.
Author Response
Dear Reviewer:
Thank you so much for your understanding, and thanks again for your helpful comments and suggestions for our manuscript.